# Review of Therapeutic Options for the Prevention of VTE in Total Joint Arthroplasty

**DOI:** 10.3390/geriatrics5010018

**Published:** 2020-03-18

**Authors:** Jordan Etscheidt, Amir Shahien, Monique Gainey, Daniel Kronenfeld, Ruijia Niu, David Freccero, Eric Smith

**Affiliations:** 1New England Baptist Hospital, Department of Orthopaedics, Boston, MA 02120, USA; jetschei@nebh.org; 2Boston Medical Center, Department of Orthopedics, Boston, MA 02118, USAruijia.niu@bmc.org (R.N.); david.freccero@bmc.org (D.F.); 3Boston University School of Public Health, Boston, MA 02118, USA; gaineym@bu.edu; 4Boston University School of Medicine, Boston, MA 02118, USA; dskronen@bu.edu

**Keywords:** total joint arthroplasty, VTE, prophylaxis, novel oral anticoagulation (NOAC)

## Abstract

Hip and knee arthroplasty patients are at high risk of perioperative venous thromboembolic events (VTE). VTE has been well studied in this population and it is recommended that total joint arthroplasty recipients receive chemoprophylactic anticoagulation due to risk factors inherent to the surgical intervention. There are few concise resources for the orthopedic surgeon that summarize data regarding post-operative anticoagulation in the context of currently available therapeutic options and perioperative standards of practice. The periodic reexamination of literature is essential as conclusions drawn from studies predating perioperative protocols that include early mobilization and sequential compression devices as standards of practice in total joint arthroplasty are no longer generalizable to modern-day practice. We reviewed a large number of recently published research studies related to post-operative anticoagulation in total joint arthroplasty populations that received a high Level of Evidence grade. Current literature supports the use of oral aspirin regimens in place of more aggressive anticoagulants, particularly among low risk patients. Oral aspirin regimens appear to have the additional benefit of lower rates of bleeding and wound complications. Less consensus exists among high risk patients and more potent anticoagulants may be indicated. However, available evidence does not demonstrate clear superiority among current options, all of which may place patients at a higher risk of bleeding and wound complications. In this situation, chemoprophylactic selection should reflect specific patient needs and characteristics.

## 1. Introduction

Hip and knee arthroplasty patients are at high risk of venous thromboembolic complications. These disorders have been well studied in this population and it is accepted that total joint arthroplasty recipients require chemical anticoagulation due to risk factors for venous thromboembolism (VTE) inherent to the surgical intervention. Accordingly, there have been numerous studies examining the utility and efficacy of post-operative VTE chemoprophylaxis following surgery; however, there are few clinically relevant, concise resources for the orthopedic surgeon that summarize data regarding post-operative anticoagulation in the context of available therapeutic options and current perioperative standards of practice. Periodic reexamination of literature is essential as conclusions drawn from studies predating perioperative protocols that include early mobilization and sequential compression devices as standards of practice in total joint arthroplasty are no longer generalizable to modern-day practice. The purpose of this review is to provide a thorough and concise, albeit non-exhaustive review of clinically relevant literature regarding VTE prophylaxis in hip and knee arthroplasty. This was achieved by highlighting recent literature with the highest levels of evidence available. Accordingly, our authors sought to review known mechanisms of action and highlight recent literature—outlining both advantages and considerations of the most commonly encountered and clinically relevant chemical anticoagulation agents.

### 1.1. General Principles

VTE is the most frequent postoperative complication in general orthopedic surgery, resulting in significant morbidity and mortality [1]. While VTE can develop after any major surgery, the incidence of VTE among orthopedic patients is substantially higher, particularly among recipients of hip and knee arthroplasty [2,3]. Underlining the importance of prevention and prophylaxis, the mortality of pulmonary embolism (PE) following diagnosis may be as high as 30%, particularly among the geriatric population [4]. Without prophylaxis, the rate of deep venous thrombosis (DVT), both asymptomatic and symptomatic, for patients undergoing hip and knee arthroplasty may be as high as 84% [3].

The potential mechanisms contributing to a postoperative hypercoagulable state following hip and knee arthroplasty are myriad. Physical manipulation of the limb, bony preparation and soft tissue manipulation may precipitate local endothelial wall trauma. Reduced post-operative mobility may contribute to venous stasis. In addition, local tissue injury may lead to an inflammatory cascade causing a temporary hypercoagulable state. These events, as described originally by Virchow, create an environment that promotes VTE [5,6].

Newly available chemoprophylactic agents range in potency, efficacy, and safety profile. The most popular prophylactic agents include low molecular weight heparin (LMWH), aspirin, vitamin K antagonists, Factor Xa inhibitors and direct thrombin inhibitors. With the exception of aspirin, each of these agents act on specific factors within the coagulation cascade. In this clinical commentary, we will review commonly utilized chemoprophylactic agents and review studies examining these agents following implementation of modern post-operative protocols, including immediate mobilization and use of pneumatic compression devices. Additionally, we seek to create a consolidated review of common anticoagulation agents with which the orthopedic surgeon should be familiar and make some general recommendations regarding their use in the hip and knee arthroplasty population.

### 1.2. Mechanisms for Intervention

Each step and mediator within the coagulation cascade serves as a potential site for pharmacologic intervention to inhibit clot formation (Figure 1 and Figure 2). This rationale is the basis for the majority of thromboprophylactic medications described below. Direct Xa inhibitors, indirect Xa inhibitors and direct thrombin inhibitors are collectively known as novel oral anticoagulants (NOAC). NOACs offer some relative advantages as targeted agents when compared to broader acting agents such as vitamin K antagonists and aspirin. LMWH will also be included in the discussion of NOACs in this section, since its mechanism of action involves a common factor with NOACs (Factor Xa). NOACs may have fewer strong drug interactions, shorter half-lives, rapid onsets of action and they lack the requirement for laboratory monitoring. Limitations of NOACs include limited use in women who are pregnant as well as a higher cost than vitamin K antagonists and aspirin [7]. 

### 1.3. Factor Xa Inhibitors

Factor Xa inhibitors bind either indirectly or directly to Factor Xa, preventing the formation of thrombin (Figure 3). Direct Factor Xa inhibitors, such as rivaroxaban (Xarelto) and apixaban (Eliquis), bind with high affinity to Factor Xa. Conversely, indirect Factor Xa inhibitors hinder thrombin formation by binding to antithrombin (AT), a serine protease inhibitor that inactivates several enzymes throughout the coagulation cascade. The binding of AT to an indirect Factor Xa inhibitor, such as fondaparinux (Arixtra), evokes a conformational change at the active site of AT, increasing its affinity to Factor Xa [8]. 

### 1.4. Direct Thrombin Inhibitors

Direct thrombin inhibitors bind directly to thrombin and block its interaction between substrates (Figure 4). A thrombin molecule has three domains for a substrate to bind to: the active site and two exosites. Exosite 1 is the location for fibrin and orients peptides into the active site; whereas, exosite 2 is primarily the heparin binding domain. Several drugs within this class of inhibitors selectively bind to one or two of these domains. Bivalent drugs bind to both the active site and exosite 1, while univalent drugs, like dabigatran (Pradaxa) bind only to the active site, disrupting the substrate thrombin interaction [9,10].

### 1.5. Low Molecular Weight Heparin

Low molecular weight heparin is a derivative of naturally occurring heparin, an anticoagulant produced by anti-inflammatory cells. LMWH, such as enoxaparin (Lovenox) and dalteparin (Fragmin), acts similarly to indirect Factor Xa inhibitors by binding to and inducing a conformational change in AT (Figure 5). This complex increases AT affinity for Factor Xa, thereby preventing the formation of thrombin from prothrombin. Unlike AT activated by heparin, AT activated by LMWH only inhibits Factor Xa rather than both thrombin and Factor Xa [9,11,12].

### 1.6. Vitamin K Antagonist

Vitamin K participates in several biochemical pathways including the coagulation cascade (Figure 6). This cofactor is required for various oxidation–reduction reactions forming vitamin K epoxide and carboxylation of glutamate residues within coagulation factors. Many coagulation factors within the cascade, including Factors II, VII, IX and X, require γ-carboxylation in order to carry out their function. Vitamin K antagonists exert their effects by interfering with the cyclic interconversion of vitamin K and vitamin K epoxide. Antagonists like warfarin (Coumadin) prevent the regeneration of vitamin K from vitamin K epoxide thus hindering the γ-carboxylation of coagulation factors necessary for the cascade to proceed [13,14].

### 1.7. Aspirin

Unlike the above-mentioned anticoagulation agents, aspirin does not manipulate a step in the coagulation cascade. Aspirin irreversibly inhibits the aggregation of platelets by acetylating a serine residue of cyclooxygenase-1 (COX-1) (Figure 7). COX-1 is essential in the arachidonic acid metabolism pathway. This enzyme catalyzes the conversion of arachidonic acid to prostaglandin H2 and the subsequent production of thromboxanes and prostaglandins. Acetylation of the serine residue within COX-1 inactivates the enzyme and prevents substrates from accessing its catalytic site. As a result, platelets are no longer able to aggregate. This antiplatelet effect lasts for the lifespan of the platelet, approximately 7 to 10 days [3,11].

### 1.8. Summary of Mechanisms

Low Molecular Weight Heparin acts similar to indirect Factor Xa inhibitors by binding to and inducing a conformational change in anti-thrombin. This complex increases anti-thrombin affinity for Factor Xa, thereby preventing the formation of thrombin from prothrombin.Factor Xa inhibitors ultimately prevent Factor Xa driven conversion of prothrombin to thrombin. Direct Factor Xa inhibitors include rivaroxaban (Xarelto) and apixaban (Eliquis). Fondaparinux (Arixtra) is an indirect factor Xa inhibitor.Direct thrombin inhibitors bind directly to thrombin, inhibiting fibrinogen conversion into fibrin.Vitamin K antagonists exert their effects by interfering with the cyclic interconversion of vitamin K and vitamin K epoxide. Antagonists like warfarin (Coumadin) prevent the regeneration of vitamin K from vitamin K epoxide and hinder the γ-carboxylation of coagulation factors necessary for the cascade.Aspirin impedes the formation of thromboxane and platelet aggregation by inhibiting the cyclooxygenase enzyme.

## 2. Materials and Methods

### 2.1. Search Strategy

Electronic searches were performed using PubMed in July 2017. Searches were performed using keywords “aspirin” OR rivaroxaban” OR “apixaban” OR “LMWH” OR “enoxaparin” OR “dabigatran” OR “warfarin” OR “thromboprophylaxis,” in combination with “knee” OR “hip” OR “joint” OR “knee replacement” or “hip replacement” OR “joint replacement” OR “knee arthroplasty” OR “hip arthroplasty” OR “orthopedics” in combination with “venous thromboembolism” and “prevention and control.” The reference lists from articles found with this method were used to identify other relevant studies.

### 2.2. Selection Criteria

Studies eligible for inclusion in the review met criteria including: English language studies, published within the last 7 years, from orthopedic focused journals (i.e., The Journal of Arthroplasty, The Journal of Bone and Joint Surgery), reported symptomatic venous thromboembolism (VTE), and reported adverse events such as bleeding complications.

### 2.3. Analysis

Studies selected for inclusion were reviewed for their rates of VTE and their direct comparison between treatment methods. Based on that, hierarchies of effectiveness and safety concerns were generated. Level of evidence grade for selected studies is summarized in Appendix A.

## 3. Results

### 3.1. LMWH Compared to Factor Xa Inhibitors

#### 3.1.1. Enoxaparin and Apixaban

Raskob et al. conducted a pooled analysis of two double-blind trials, the ADVANCE-2 and ADVANCE-3 trials, in 2012 which included 8464 total patients in a direct comparison of LMWH to Factor Xa Inhibitors. The treatment arms included one group which received apixaban 2.5 mg twice daily (plus placebo injection) starting 12 to 24 h after operation, and the other which received enoxaparin subcutaneously once daily (and placebo tablets) starting 12 h pre-operatively.

There were statistically fewer VTE events in the apixaban group when compared with the enoxaparin cohort (*p* < 0.001) without a significant increase in bleeding events, risk difference 0.6% (95% confidence interval (CI) −1.5 to 0.3) [15]. Authors concluded that apixaban 2.5 mg twice daily is more effective than enoxaparin 40 mg once daily without increased bleeding.

#### 3.1.2. Enoxaparin and Rivaroxaban

Erikson et al. compared once-daily dose of rivaroxaban 10 mg, an oral, direct Factor Xa inhibitor, with enoxaparin 40 mg subcutaneously once daily in a pooled analysis of three separate studies for patients undergoing both elective total hip arthroplasty (THA) and total knee arthroplasty (TKA) (n = 9581). Rivaroxaban significantly reduced the incidence of both VTE events and all-cause mortality at the end of the treatment regimens, odds ratio 0.38; 95% CI 0.22 to 0.62; *p* < 0.001. There was no difference in bleeding between the two regimens. The authors concluded that rivaroxaban started six to eight hours after surgery was more effective than enoxaparin started the previous evening in preventing symptomatic venous thromboembolism and all-cause mortality, without increasing major bleeding [16]. A recent meta-analysis including forty-five randomized controlled trials of 56,730 patients by Suen et al. found similar effectiveness of VTE prophylaxis with enoxaparin, rivaroxaban and apixaban when compared to a warfarin control, with a trend towards increased efficacy of VTE prophylaxis with the use of enoxaparin. Comparison of bleeding events in this 2017 study revealed a 2.32 relative risk (RR), 95% CI, 1.40–3.85 of increased bleeding with control and 1.54 RR, 95% CI 1.23–1.94 when compared to warfarin. Authors concluded that LMWH increased the risk of surgical site bleeding compared with warfarin and dabigatran. The risk of surgical site bleeding was similar with LMWH and rivaroxaban [17]. 

### 3.2. LMWH Compared to Direct Thrombin Inhibitors

#### Enoxaparin and Dabigatran

Ginsberg et al. conducted a double-blind, randomized trial in patients receiving unilateral total knee arthroplasty. In the three treatment arms of this study, patients received either oral dabigatran etexilate 220 or 150 mg once daily, or enoxaparin 30 mg subcutaneous, twice daily. In this study of 1896 patients, the RE-MOBILIZE research consortium found dabigatran to be significantly less effective than enoxaparin (VTE events: 31%, *p* = 0.02 vs. enoxaparin; 34%, *p* = 0.001 vs. enoxaparin, and 25%, respectively) for the prevention of VTE events. Bleeding events were found to be similar in both regimens. Authors concluded that dabigatran, although effective compared to once-daily enoxaparin, showed inferior efficacy to the twice-daily North American enoxaparin regimen [18].

### 3.3. LMWH Compared to Warfarin

#### 3.3.1. Enoxaparin and Warfarin

No high level of evidence, randomized control trials directly comparing enoxaparin and warfarin have been published since 2001. In 2001, Fitzgerald et al. conducted a multicenter, parallel group, randomized control trial including 349 patients and treated each intervention arm with either enoxaparin, 30mg aspirin twice daily (BID), or warfarin, goal international ratio (INR) 2–3, immediately followingTKA [19]. VTE developed in significantly fewer (*p* = 0.0001) enoxaparin-treated patients and the enoxaparin-treated patients also had a significantly lower prevalence of proximal deep-vein thrombosis (*p* = 0.002). There was no significant difference (*p* = 0.15) between groups with regard to the occurrence of major hemorrhagic complications; however, the rate of overall hemorrhagic complications was higher in the enoxaparin group. More recently, a 2010 Cochrane review by Salazar et al. examined direct thrombin inhibitors versus vitamin K antagonists and LWMH for prevention of VTE following total hip or knee replacement. The review of 14 studies involving 21,642 patients concluded that direct thrombin inhibitors are as effective in the prevention of major venous thromboembolism in THA or TKA as LMWH and vitamin K antagonists. However, they show higher all-cause mortality odds ratio (OR) 2.06 (95% CI 1.10 to 3.87) and result in more bleeding events OR 1.40 (95% CI 1.06, 1.85) than LMWH [20].

#### 3.3.2. Dalteparin and Warfarin

Gillette et al. examined the rate of symptomatic VTE events following three treatment regimens including aspirin 325 mg, warfarin (target INR, 1.8–2.2), and dalteparin in a retrospective review of 2046 patients who underwent either primary TKA or THA. These patients also received tranexamic acid intraoperatively. They reported a low complication rate and no difference in the rates of DVT and PE between the groups. For aspirin, warfarin and dalteparin, the rates of symptomatic DVT (0.35%, 0.15% and 0.52%, respectively) and nonfatal PE were similar (0.17%, 0.43% and 0.26%, respectively) [21]. Authors concluded that when using TXA, less aggressive thromboprophylactic regimens such as aspirin alone and dose-adjusted warfarin are appropriate.

### 3.4. Aspirin Compared to Low Molecular Weight Heparin (LMWH)

A 2016 systematic review conducted by Wilson et al. examined aspirin as a thromboprophylactic agent in THA and TKA patients. A review of the thirteen papers included demonstrated rates of asymptomatic DVT in TKA may be reduced with rivaroxaban when compared with aspirin, but insufficient evidence exists to demonstrate an effect on incidence of symptomatic DVT. Compared with aspirin there is evidence of more wound complications following THA and TKA with dabigatran and in TKA with rivaroxaban. Authors suggest aspirin as a sufficient alternative to other thromboprophylactic agents following THA and TKA [22].

A 2016 retrospective cohort study by Nielen et al. compared rates of venous thromboembolism (VTE), gastro-intestinal (GI) bleeding and mortality events in 3261 TKA recipients and 4016 THA recipients that were prescribed either NOACs only, LMWHs only, or aspirin only. Both TKA and THA recipients on LMWH were found to have increased risk of VTE, Hazard Ratio (HR) = 17.2 (CI 6.9–43.0) and HR = 39.5 (CI 18.0–87.0); GI bleeding, HR = 20.9 (CI 1.9–232.3) and HR = 2.0 (CI 0.2–17.2); and mortality HR = 4.3 (CI 1.7–12.4) and HR = 4.0 (CI 2.4–6.7) compared to NOACs and aspirin, respectively. NOAC use was associated with an increased risk of GI bleeding in patients undergoing THR surgery [23]. Authors concluded an increased risk of VTE, GI bleeding and all-cause mortality with the use of LMWHs compared with aspirin.

Anderson et al. conducted a multicenter, randomized control trial of 778 patients undergoing elective unilateral THA. All patients received an initial 10 days of dalteparin prophylaxis and were then randomized into 28 days of dalteparin or aspirin. Clinically significant bleeding occurred in five patients (1.3%) receiving dalteparin and two (0.5%) receiving aspirin. Although the study was stopped prematurely due to recruitment difficulties, the authors’ conclusion suggested extended 28 day prophylaxis with aspirin was as safe as, and non-inferior to, dalteparin in preventing VTE after total hip arthroplasty [24].

### 3.5. Aspirin Compared to Factor Xa Inhibitors (Rivaroxaban as an Example)

In addition to the Wilson et al. systematic review mentioned previously, Weitz et al. conducted an industry-sponsored randomized, double-blind phase 3 study including 3396 patients with VTE who were treated for a mean of 351 days [22]. This intention to treat study examined the rate of symptomatic recurrent fatal or nonfatal venous thromboembolism in the context of major bleeding. The study suggests that recurrent event of VTE was significantly lower with 20 mg rivaroxaban vs. 325 mg aspirin, HR 0.34; 95% CI, 0.20 to 0.59; as well as for 10 mg of rivaroxaban vs. aspirin, 0.26; 95% CI, 0.14 to 0.47; *p* < 0.001 for both comparisons [25]. Rates of major bleeding were 0.5% in the group receiving 20 mg of rivaroxaban, 0.4% in the group receiving 10 mg of rivaroxaban, and 0.3% in the aspirin group. Authors concluded the risk of a recurrent event was significantly lower with rivaroxaban at either a treatment dose (20 mg) or a prophylactic dose (10 mg) than with aspirin, without a significant increase in bleeding rates.

### 3.6. Aspirin Compared to Direct Thrombin Inhibitors (Dabigatran as an Example)

A 2014 retrospective review of 1728 patients receiving primary total joint arthroplasty before and after the implementation of dabigatran as thromboprophylactic agent revealed a significant increase in post-operative wound leakage (20% with dabigatran vs. 5% with a multimodal regimen including LMWH and extended use aspirin; *p* < 0.001). There was also a significant increase in duration of hospital stay. Bloch et al. demonstrated that the multimodal VTE prophylactic regimen of enoxaparin inpatient and extended use of aspirin after discharge resulted in significantly lower rates of symptomatic VTE, wound leakage and length of stay in post-operative total knee and total hip arthroplasty patients and resulted in the discontinuation of dabigatran for thromboprophylaxis [26].

### 3.7. Aspirin Compared to Warfarin

A 2012 study including 696 consecutive total joint arthroplasty patients stratified 281 patients according to PE risk stratification per American Academy of Orthopaedic Surgeons guidelines. One hundred and fifty-two standard-risk patients received aspirin, and 129 elevated-risk patients received warfarin. A comparator group of 415 patients received American College of Chest Physicians-recommended warfarin without PE risk stratification. The rate of symptomatic PE and venous thromboembolism among standard-risk group patients receiving aspirin was greater than the warfarin group (4.6% vs. 0.7% and 7.9% vs. 1.2%, respectively) [27]. Contrarily, Huang et al. published a 2016 retrospective study with 30,270 patients suggesting aspirin as effective as and safer than warfarin for VTE prophylaxis after hip and knee arthroplasty, even in patients at higher risk of VTE [28]. The incidences of VTE, prosthetic joint infectionand mortality were significantly higher in patients receiving warfarin compared to aspirin. In multivariate analysis, warfarin was an independent risk factor for VTE, PJI and mortality in the higher risk VTE patients (*p* < 0.001). There was no significant difference in gastrointestinal complications between groups.

### 3.8. Aspirin High Dose Compared to Aspirin Low Dose

A prospective, crossover study from the Rothman Institute compared the effects of 81 mg BID to 325 mg BID. This study showed no significant difference (*p* > 0.05) in the rates of VTE and a no significant difference in gastrointestinal complications, acute peri-prosthetic joint infection and 90-day mortality rate between the two groups [29].

### 3.9. Emerging Trends

Previous trends toward aggressive perioperative anticoagulation for VTE prophylaxis and the concomitant exponential increase in total joint arthroplasty has led to the increased awareness of post-operative bleeding and wound drainage complications [30]. Over the past decade there has been a movement toward the use of aspirin as the primary agent for VTE prophylaxis following total joint arthroplasty. Multiple recent studies have validated the use of aspirin in the setting of both primary and revision total joint arthroplasty [3,29,31,32]. In a retrospective review of 30,499 unilateral TKA patients receiving either 325 mg aspirin daily, low-molecular-weight heparin (enoxaparin 40–60 mg daily), synthetic pentasaccharide factor Xa inhibitors (fondaparinux 2.5 mg daily), or vitamin K antagonist (warfarin, all doses), Cafri et al. reported no difference with aspirin when compared to the other agents in the prevention of pulmonary embolism, deep vein thrombosis or venous thromboembolism. The authors reported no “evidence for decreased effectiveness or increased safety with use of aspirin, but enoxaparin had comparable safety to aspirin for bleeding and fondaparinux had comparable safety to aspirin for wound complications” [33].

Additional studies suggest that aspirin is a viable therapeutic option for VTE prophylaxis in patients stratified into groups with high risk of VTE events following total joint arthroplasty [28,34]. Also contributing to the appeal of aspirin as a first-line agent for chemical VTE prophylaxis is its low cost relative to alternative therapeutic options and its efficacy and safety profile when combined with other therapeutic agents such as intraoperative tranexamic acid [35,36]. It is appropriate to start aspirin post-operatively with the simultaneous resumption of pre-operative, chronic anticoagulation regimens with good efficacy given its mechanism of action outside of the traditional coagulation cascade.

## 4. Discussion

Increased understanding of the coagulation cascade has led to advancements in the development of anticoagulation agents and increased pharmacologic options. Given the high satisfaction rate and low rate of revision following total joint arthroplasty utilizing contemporary surgical technique and improved biomaterials, protection of the post-operative surgical wound and related complications are of paramount importance. Given the emerging trends of abbreviated inpatient stay following TJA and the increasing percentage of total joint arthroplasty performed in the outpatient setting, the search for reliable chemical anticoagulation agents that reduce VTE events while minimizing post-operative bleeding and wound complications is of great interest [37]. Not all new agents have been able to achieve these objectives consistently; many have been associated with an increased incidence of wound leakage and increased length of stay after total joint replacement [26]. Continued exploration of novel agents as they become commercially available is essential.

### 4.1. Reconciling Current Guidelines

There has been a struggle for consensus regarding thromboprophylactic guidelines as various societal and specialty specific guidelines have given conflicting recommendations. The argument of Parvizi et al. that American College of Chest Physicians (ACCP) guidelines were overly aggressive in the dosage and agent choice following total joint arthroplasty and did not properly emphasize limiting post-operative bleeding complications and wound protection is still of clinical concern today [30,38,39]. Historically, a discrepancy has existed between medical societies in measuring the efficacy of VTE prophylaxis. The 9th Edition ACCP guidelines published in February 2012 broadly recommends use of multiple agents including enoxaprin, apixaban, rivaroxaban or fondipariux to reduce the incidence of VTE [40]. ACCP guidelines endorse more frequent, higher doses of enoxaparin in an attempt to prevent PE. The ACCP guidelines, however, include both asymptomatic and symptomatic DVT detected by venography as a measure of the efficacy of thromboprophylaxis, whereas the AAOS has rejected asymptomatic DVT as a valid outcome as the council considers the association between DVT and PE to be unproven [41].

The American Academy of Orthopedic Surgeons has one “strong” recommendation concerning VTE anticoagulation: recommending against the use of routine post-operative duplex ultrasonography screening of patients who undergo elective hip or knee arthroplasty. Citing lack of definitive evidence in this area, the academy currently endorses several “consensus recommendations” which include early mobilization and mild anticoagulation—without specification of agent—paired with mechanical compression devices [42]. Moderate recommendations of the society include discontinuing antiplatelet agents (e.g., aspirin, clopidogrel) before undergoing elective hip or knee arthroplasty and the use of neuraxial (such as intrathecal, epidural and spinal) anesthesia for patients undergoing elective hip or knee arthroplasty to help limit blood loss, even though evidence suggests that neuraxial anesthesia does not affect the occurrence of venous thromboembolic disease [42,43,44].

### 4.2. Authors Recommendations

Clearly, significant conflict remains regarding optimal thromboprophylactic selection, dosing and safety. This may be related to significant practice, population and study design heterogeneity that exists within the body of literature that limits the ability to create consensus recommendations. Substantial further work is required to clarify optimal usage guidelines for hip and knee arthroplasty patients, in addition to more clearly defining the risk profiles of these various agents. However, in general, we can make several recommendations based upon this review.

Among patients not considered to be at “high risk” for postoperative VTE, the authors recommend the use of oral aspirin rather than more potent anticoagulants. Although some conflicting evidence exists, there appears to be emerging consensus that aspirin is non-inferior to more potent anticoagulants for symptomatic VTE prophylaxis and carries significantly lower risk of bleeding and wound complications. Aspirin dosing remains an area of controversy, but limited evidence supports the use of low-dose regimens. Further work is required to define an optimal duration of therapy, however, given the low cost and low risk of aspirin thromboprophylaxis, we recommend a duration of therapy of at least 4 weeks.

For patients considered “high risk” not already receiving chronic anticoagulation, our recommendations are less decisive. Significant conflict exists within the literature regarding efficacy, dosing and safety of the myriad agents now available to practitioners. Furthermore, studies focused on high risk populations are lacking, and there is controversy regarding the optimal methodology for VTE risk stratification, which remains outside the scope of this review. Recognizing these limitations, among “high risk” patients, the authors recommend the use of a more potent anticoagulant, understanding that there may be a tradeoff between improved VTE prophylactic efficacy and increased risk of bleeding and wound complications. Insufficient evidence exists for us to make specific agent, dosing or duration of therapy recommendations. Instead, we would advise that practitioners take patient comorbidities, postoperative activity levels and individual characteristics into account when selecting optimal chemoprophylactic regimens.

## 5. Conclusions

Although the above guidelines reflect our institutional preferences for best practices, they are not representative of current academic societal guidelines. This is primarily a reflection of the significant limitations that exist within the current body of literature. While there is consensus that chemical VTE prophylaxis is necessary, the conversation continues regarding best therapeutic options for post-operative VTE prophylaxis [45]. A 2008 survey of American Association of Hip and Knee Surgeons (AAHKS) membership, published by Markel et al. in 2010, explored current VTE protocols for lower-extremity total joint surgery. Fifty-three percent of respondents reported a change in VTE-related practices over the previous five years. Seventy-four percent of responding surgeons reported that their hospital recognized the ACCP guidelines, however 68% believed AAOS guidelines to be more clinically relevant to daily practice. At the time of the survey, warfarin was most used in hospital practice [46]. Ultimately, working knowledge of available thromboprophylactic agents is essential for the orthopedic surgeon as optimal agents will likely become patient specific as the literature continues to evolve.

## Figures and Tables

**Figure 1 geriatrics-05-00018-f001:**
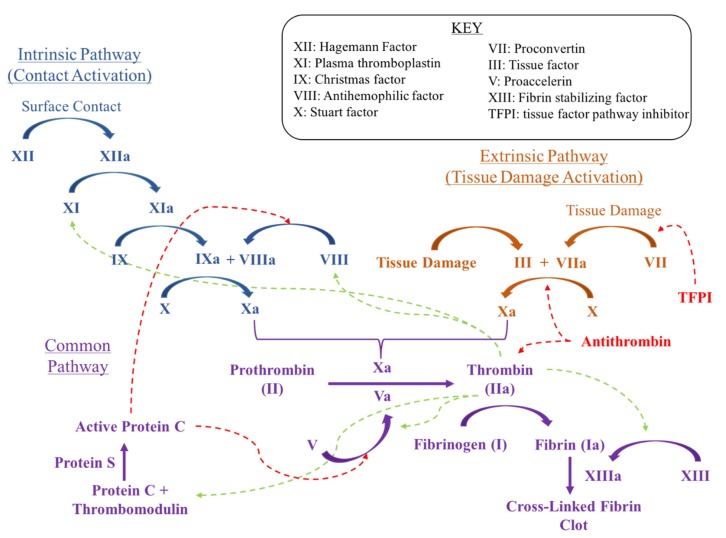
Overview of the coagulation pathway.

**Figure 2 geriatrics-05-00018-f002:**
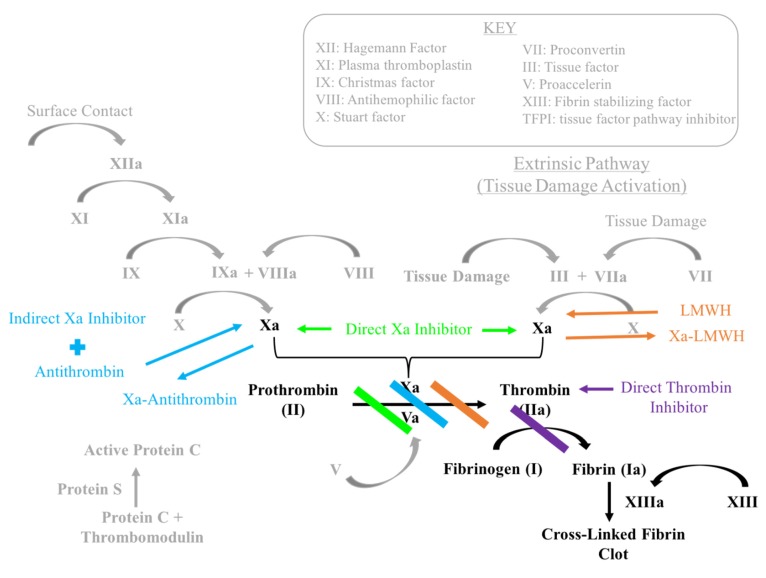
Overview of the coagulation pathway.

**Figure 3 geriatrics-05-00018-f003:**
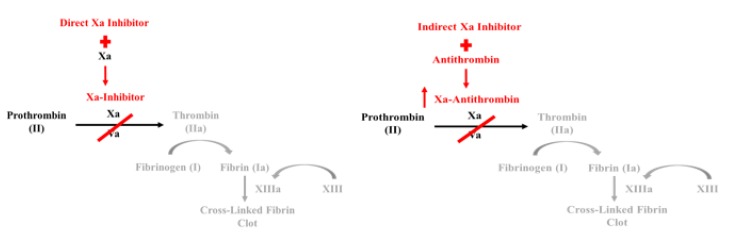
Mechanism of Action for Factor Xa Inhibitors.

**Figure 4 geriatrics-05-00018-f004:**
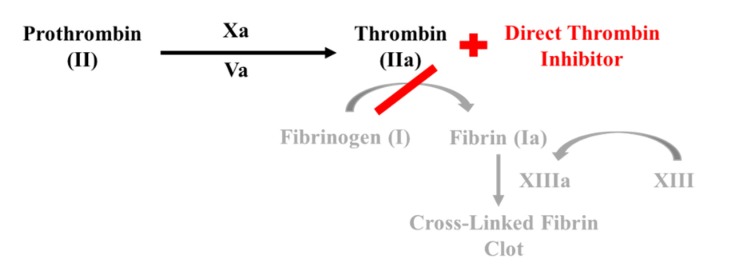
Mechanism of action for direct thrombin inhibitor.

**Figure 5 geriatrics-05-00018-f005:**
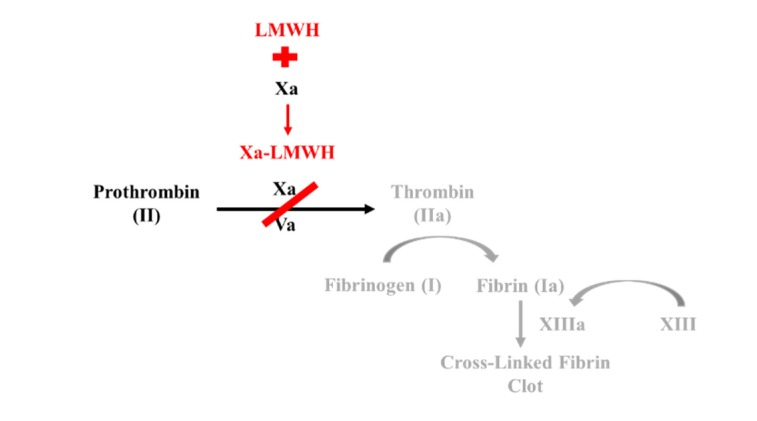
Mechanism of Action for low molecular weight heparin.

**Figure 6 geriatrics-05-00018-f006:**
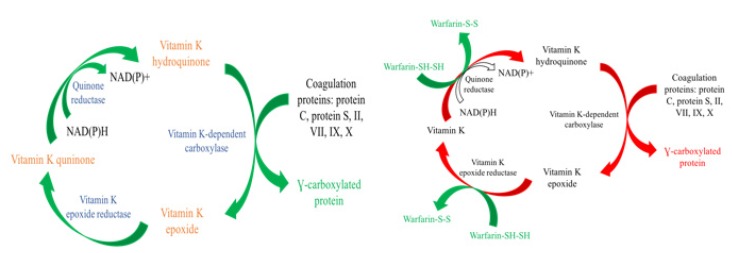
Normal Vitamin K cycle vs. Vitamin K antagonist. Compared to the normal pathway on the left, warfarin is reduced by Vitamin K epoxide reductase and quinone reductase preventing the regeneration of Vitamin K and subsequent carboxylation of coagulation factors.

**Figure 7 geriatrics-05-00018-f007:**
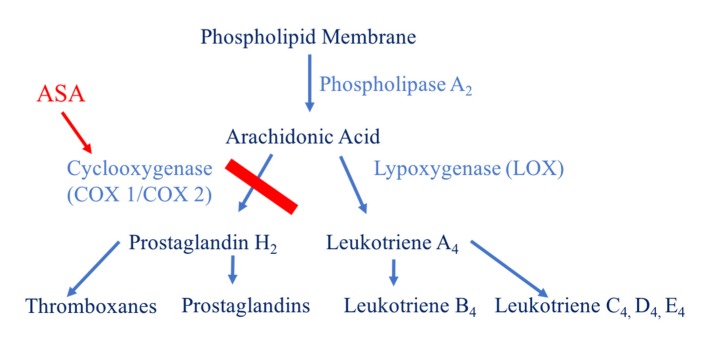
Mechanism of action of aspirin (ASA). The binding of aspirin to COX-1 or COX-2 prevents the formation of prostaglandin H2 and subsequently thromboxane and prostaglandin, affecting platelet aggregation.

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
