# Peer review of "Review of Therapeutic Options for the Prevention of VTE in Total Joint Arthroplasty"

_geriatrics, 2020, doi:10.3390/geriatrics5010018_

Round 1

Reviewer 1 Report

1 owing to the new development of anti-coagulants, as authors mentioned in this article, the revised guideline for daily practice should be established. 

2 authors collected most of the important studies and well described the importance of these studies 

3 however, it would be better if authors could possibly provide the evidence level of each literature by using critical appraisal tools such as CASP, Oxford CEBM or Cochrane RoB.

Author Response

We appreciate the appraisal by Reviewer 1 on the clinical significance and comprehension of our review study. In response to the 3rd point in the reviewer’s comments, we are not conducting a systematic review; therefore, we don’t think a formal assessment of risk of bias is necessary. Instead, we added an appendix table to demonstrate the level of evidence for all the articles we included.

Reviewer 2 Report

General comments

The authors provide a review/clinical commentary of existing literature regarding venous thromboembolism prophylaxis in total joint arthroplasty.

Topic is very relevant and will be a valuable resource to surgeons and healthcare providers. Some comments worth considering.

The authors indicate this is a clinical commentary and reviewed the available evidence but were unable to provide any concrete conclusions on which were the most effective thrombprophylactic agents. Please consider doing this by weighing the overall evidence and the balance between benefits and adverse effects.

Total joint arthroplasty is quite a broad term. Whereas there are established guidelines on the use of thromboprophylaxis in patients undergoing hip and knee replacements, those for shoulder and elbow replacements are not clearly defined because only few of these replacements are ever carried out every year. The authors will need to be specify which joints the review focussed on and not generalise it to all joints. Please consider revising in all aspects of the draft. The authors should consider briefly reviewing the evidence for shoulder and elbow replacement too given the limited literature in this area.

Specific comments

Abstract

It will be useful to provide a brief summary of the main findings of the review. Currently, the abstract only focuses on the background, rationale, and methods. This doesn’t provide much information to readers. A couple of sentences on the relevant findings will also be useful. Which agents do the authors recommend given the available evidence? “Orthopedic patients are at high risk for venous thromboembolic events.” This is quite a vague time. I don’t think all orthopaedic patients are at high risk of VTE. A patient undergoing surgery in the wrist is not at high risk of VTE compared to one undergoing hip replacement. Please consider revising and in other relevant sections. “Our authors reviewed a large number of recently….” The use of “our” sounds quite impersonal. Why not use “We”? Please revise in other sections

Introduction

“Accordingly, there have been numerous studies examining the utility and efficacy of post-operative VTE prophylaxis following total joint arthroplasty;” Please provide some relevant references. “The author’s…” Was only one author involved in this review? Please revise

General Principles

“VTE is the most frequent postoperative complication in orthopedic surgery…” Orthopaedic surgery is a very broad speciality and the incidence of VTE is not the same across all areas of orthopaedics. Consider revising to arthroplasty if this is what the authors mean. Please revise throughout the manuscript as used in several instances There are several figures, but no links to these figures have been provided in the text

Materials and Methods

Though a clinical commentary, it will be useful to provide some brief information on the databases employed and some of the search terms employed.

The Authors should consider dedicating a section to the safety profile of these therapeutic options.

Author Response

We focused our review on the total joint arthroplasty in lower extremity; we also realized the use of “orthopaedic surgery” is too broad and vague in our context. Therefore, to make it specific, we changed all relevant texts, such as “total joint arthroplasty” and “orthopaedic surgery”, in the manuscript to be “hip and knee arthroplasty”.

We added our conclusion on the literature review in the abstract (last 3 sentences).  To elaborate on this conclusion, we also added a section on our opinions after this review (Section 4.2 Authors Recommendations).

We demonstrated the safety profile of each therapeutic options in their respective sections.

Finally, we added a method section.

Reviewer 3 Report

I appreciate the authors' remarkable effort to write such a concise paper on a vastly researched topic of VTE. It is well-written and is very interesting to read. The topic is of a high-interest among Orthopaedic surgeons of all backgrounds and interests.

I have 2 suggestions:

Methodology section requires some more elaboration i.e. to include which databases were searched, what search terminologies were used and how many articles were identified. Despite of not being a systematic review, adding this information will be useful.

Secondly, just to clarify regarding the pictures of clotting cascade; are these produced by the authors themselves or taken from any other source? in case of the latter, an appropriate permission/reference will need to be added.

Author Response

We thank Reviewer 3 for the recognition on the clinical implications of our review article. Methodology section was added to elaborate on our searching terms, databases, selection criteria, and analysis of literature data. All the figures are created originally by us.

Round 2

Reviewer 2 Report

Authors have addressed all comments.